# Environmental Determinants of Early Childhood Caries: A Narrative Synthesis of Observational Evidence and Implications for Global Policy

**DOI:** 10.3390/dj13110484

**Published:** 2025-10-22

**Authors:** Moréniké Oluwátóyìn Foláyan, Robert J. Schroth, Olubukola Olatosi, Maha El Tantawi

**Affiliations:** 1Early Childhood Caries Advocacy Group, University of Manitoba, Winnipeg, MB R3E 0W2, Canada; 2AFRONE Network, Faculty of Dentistry, Alexandria University, Alexandria 21527, Egypt; 3Department of Child Dental Health, Obafemi Awolowo University, Ile-Ife 22005, Nigeria; 4Oral Health Initiative, Nigerian Institute of Medical Research, Yaba, Lagos 100001, Nigeria; 5Dr. Gerald Niznick College of Dentistry, University of Manitoba, Winnipeg, MB R3E 0W2, Canada; 6Department of Child Dental Health, University of Lagos, Lagos 101017, Nigeria; 7Department of Pediatric Dentistry and Dental Public Health, Faculty of Dentistry, Alexandria University, Alexandria 21527, Egypt

**Keywords:** planetary health, perfluoroalkyl acids, ecovitality, exposome, universal health coverage, Sustainable Development Goals, One Health approach, disability-adjusted life years

## Abstract

Early childhood caries (ECC) remains a significant global health challenge, disproportionately affecting marginalized populations. While traditional research emphasizes behavioral and biological risk factors, emerging evidence highlights the critical role of environmental determinants. This narrative synthesis aims to highlight the role of environmental determinants as a risk factor for ECC pathogenesis. Environmental toxins (e.g., lead, perfluoroalkyl acids, tobacco smoke, air pollution) disrupt enamel development, impair salivary function, and compromise immune responses, directly increasing caries susceptibility. Environmental degradation, including air pollution, reduces ultraviolet B radiation exposure, limiting endogenous vitamin D synthesis that is vital for enamel mineralization and immune regulation. These risks are compounded in low- and middle-income countries, where structural inequities, inadequate sanitation, and climate disruptions exacerbate ECC burdens. We introduce ecovitality—the resilience of ecosystems supporting human health—as a novel framework linking ecological vitality to oral health. Degraded environments limit access to fluoridated water and nutrient-dense foods while promoting sugary diets and endocrine disruptors. A One Health approach is advocated to address interconnected environmental, social, and biological determinants of the risk for ECC. Despite global reductions in tobacco use and lead exposure, the Global Burden of Disease 2021 analysis reports stagnation in ECC prevalence. This underscores the critical need for longitudinal and mechanistic studies to establish causality, quantify the contributions of environmental controls, and explore how mitigating these risks can reduce the global ECC burden. Such evidence may promote interdisciplinary action to align oral health promotion for children with the Sustainable Development Goals.

## 1. Introduction

Early childhood caries (ECC) refers to cavitated and non-cavitated decayed, filled, or missing primary teeth in children under 72 months of age, as first defined at a workshop held 28–29 April 1999 [1,2]. The 1999 Drury workshop definition established critical diagnostic criteria for research and public health surveillance [1]. More recently, the 2019 IAPD Bangkok Declaration [2] reaffirmed this clinical definition while providing a simplified lay definition (‘tooth decay in children under age 6 years’) and emphasizing the role of socioeconomic and environmental inequities as root causes. Together, these frameworks standardize ECC identification and underscore its status as a global health injustice requiring multisectoral policy action.

Defining ECC as a distinct clinical entity standardized its varied features and etiological factors, enabling consistent epidemiological surveillance, cross-population burden comparisons, and targeted public health interventions [3]. Like caries in permanent teeth, ECC arises from enamel and dentin demineralization caused by interactions between cariogenic microorganisms and dietary fermentable carbohydrates [4]. If untreated or treatment is delayed, ECC imposes substantial physical, psychological, and financial burdens on children and parents, underscoring the need for coordinated public health action [5]. Despite ECC being preventable, the prevalence has continued to rise in many vulnerable communities at the micro-, meso-, and macro-level. It is, therefore, imperative to understand the factors that drive the risk for ECC understanding, especially among the vulnerable population.

One possible risk is nutrition-related behavioural risk factors—such as wasting, underweight, and stunting. However, global trends show a decline in malnutrition between 1990 and 2019 [6]. In contrast, environmental risks, including unsafe sanitation, inadequate handwashing, and ambient particulate matter, increased over the same period, while household air pollution and unsafe water remained persistently high [7]. These environmental exposures are possible risk factors for ECC [8,9]. Their impact is shaped by gene–environment interactions applicable to NCDs [10,11].

The environmental impact on oral health is likely to be high for children who are generally more vulnerable to the harmful effects of environmental toxins due to their limited capacity for detoxification and increased biological sensitivity to environmental exposures [12]. These environmental influences may also interact with biological determinants of health to shape the risk of ECC. Dental caries risk are associated with the amount and frequency of sugar consumed [13], oral hygiene practices [14], and nutritional risk factors [15], which may be influenced by broader environmental conditions such as clean water and sanitation [16], clean energy [17], urbanization [18], waste and pollution [19], the climate [20], oceans, marine [21], and land [22] resources.

Moreover, the global demand for sugar, a primary behavioural risk factor for ECC, drives intensive agricultural practices such as sugarcane monoculture. These practices contribute to habitat fragmentation, biodiversity loss, and degradation of air, water, and soil quality [23]. For instance, sugarcane cultivation often involves deforestation, agrochemical runoff, and soil depletion, disproportionately affecting low- and middle-income countries where environmental regulations are weak [24]. This degradation exacerbates exposure to pollutants (e.g., pesticides, heavy metals) and reduces access to diverse, nutrient-rich foods [25,26], which can indirectly promote reliance on sugar-laden processed diets. Thus, the environmental cost of sugar production creates a cyclical risk: ecological damage amplifies ECC susceptibility while sustaining the very dietary habits that drive caries. Therefore, understanding how environmental determinants of ECC can exacerbate or mitigate this impact represents a strategic frontier for achieving Sustainable Development Goal (SDG) 3.1 (reducing under-5 mortality and ensuring healthy development). The SDGs provide a critical framework for understanding and addressing the multifactorial nature of ECC, as its determinants span goals related to health (SDG 3), clean water and sanitation (SDG 6), sustainable consumption and production (SDG 12), climate action (SDG 13), and reducing inequalities (SDG 10) [16,17,18,19,20,21,22].

This narrative synthesis aims to highlight the role of environmental determinants as a risk factor for ECC pathogenesis. We synthesized emerging evidence on environmental determinants of ECC as modifiable risk amplifiers that exacerbate susceptibility in vulnerable populations, as these are understudied pathways that may have implications for global policy. We also explored the intersections with the SDG, positioning ECC not just as a dental disease but as a marker of socio-environmental inequity and a barrier to achieving sustainable development. We posit that understanding the interactions between behavioural, biological, and environmental factors is essential for developing holistic interventions for ECC, and invite critical scholarly debate on the planetary health dimensions of oral disease.

## 2. Methods

This paper adopts a narrative synthesis methodology to examine how complex, multi-level environmental and social determinants shape the burden of ECC. The approach begins with clearly defining the outcome—ECC prevalence, severity, and untreated cases—and its disproportionate impact on vulnerable populations. A search of the literature for studies that examined connections between environmental exposures and ECC, either directly or through intermediate pathways, was organized thematically, and the information was synthesized. Within each theme, studies were compared to uncover relationships, patterns, and mechanisms. The analysis identified how biological, behavioural, and social mechanisms intersect, and assessed the strength and consistency of the evidence supporting each link. The evidence strength for the findings on the link between environmental exposures and ECC was categorized using established frameworks (GRADE, NTP Handbook) based on: (1) consistency of human studies, (2) dose-response relationships, (3) mechanistic plausibility, and (4) risk of bias (e.g., confounding). High-strength evidence required multiple consistent observational studies; moderate strength indicated dose-response but unresolved limitations; low strength relied primarily on mechanistic data [27]. Based on the findings, a conceptual framework was developed.

## 3. Smoked Tobacco and Early Childhood Caries

At the household level, a significant air pollutant is tobacco smoking, with a significant moderate association between passive tobacco exposure [28], secondary smoking [9,29], and caries in children. The evidence on the causal effect is stronger for ECC than for caries in the permanent dentition [30,31], and the risk is dose-related [32]. Prior studies also highlighted associations between ECC and postnatal [33,34,35,36,37,38,39] and prenatal [40,41,42] tobacco smoking, although inconsistencies in the literature warrant careful consideration, as some studies reported null associations [43,44]. Methodological limitations may contribute to this inconsistency. For example, self-reported smoking status can lead to exposure misclassification, especially for second-hand smoke, due to recall bias or underreporting [40]. In settings where sugar consumption is universally high or access to fluoride is limited, the specific impact of tobacco exposure may be obscured by these stronger confounders. Study design also plays a role; cross-sectional studies cannot establish temporal causality, in contrast to longitudinal cohort studies. In addition, studies conducted in high-income countries with strong tobacco regulations may observe weaker associations due to lower exposure intensity.

Despite these variations, the overall weight of evidence supports a causal link between tobacco smoke exposure and ECC. Tobacco smoke contributes to ECC through direct biological pathways: nicotine disrupts ameloblast function, causing hypomineralized enamel; it alters the oral microbiome (e.g., enriching *Streptococcus mutans*); and impairs salivary gland function, reducing pH and clearance of cariogenic substrates [39,45,46,47]. The use of e-cigarettes also alters the oral microbiome, increasing *Streptococcus mutans* count [48].

There is a concern as the popularity of e-cigarettes has increased [49] despite the global decline in tobacco smoking resulting from synergistic policies (taxation, bans, packaging), health system efforts (cessation programs, campaigns), and cultural shift [50]. Tobacco taxes increase the product price, reducing affordability, especially among low-income groups [51,52,53]. A 10% price increase typically reduces tobacco consumption by approximately 4% in high-income countries [54] and 5–8% in low- and middle-income countries [55,56,57]. The price increase causes a reduction in the number of cigarettes smoked by those who continue smoking [58] and delays or prevents smoking initiation in various countries. Restricting tobacco marketing can reduce smoking prevalence by 4% in the short term and 6% in the long term [59]. Laws for smoke-free public spaces have also significantly reduced secondhand smoking and denormalized smoking [60]. Furthermore, removing branding and adding health warnings decreases appeal, especially for new smokers [61]. In addition, access to nicotine replacement therapies, counseling services, and smoking quit lines significantly enhances smoking cessation [62]. Mass media campaigns have been effective in reshaping public attitudes and social norms around tobacco use [63], while legal actions and transparent regulations have helped reduce the influence of the tobacco industry [64,65]. Policy-led denormalization efforts have also contributed to the decline in the social acceptability of smoking [66]. Although the role of e-cigarettes remains contentious, their availability may have contributed to reduced smoking rates, especially in high-income countries [67,68].

With shrinking markets in the Global North, tobacco companies have intensified their marketing efforts across Africa and Southeast Asia, where policy enforcement is often weaker [69]. Smoking continues to be disproportionately prevalent among marginalized populations [70], perpetuating cycles of health inequity.

These broad trends in tobacco control may have indirect yet meaningful implications for ECC through several interconnected pathways. First, a reduction in tobacco use can lower children’s exposure to secondhand smoke. In addition, declining smoking rates may signal improvements in broader health behaviors within households, potentially fostering better oral health environments for children. Furthermore, tobacco control policies often intersect with public health and educational campaigns, which can indirectly promote greater awareness of child health issues, including oral health. In settings where maternal smoking decreases due to effective policies, there may also be fewer adverse pregnancy outcomes, such as low birth weight, that are associated with increased susceptibility to ECC [71]. Finally, tobacco-related revenues could be reallocated to health promotion and prevention programs, and in doing so, there is the potential for increased investment in child health initiatives, including oral health services and education.

## 4. Lead Poisoning and Early Childhood Caries

Lead is also a major cause of household pollution [72] and has been linked to increased prevalence and severity of ECC at elevated blood levels [73,74,75,76,77,78,79]. There are no longitudinal studies on the dose-dependent relationship between elevated blood lead levels and ECC prevalence/severity that could offer stronger causal inference. However, cross-sectional analyses reporting null associations [75,76,79,80,81,82,83,84] are likely limited by snapshot exposure assessments and residual confounding (diet, access to care). The paradoxical finding by Kim et al. [85] of elevated caries at low lead levels (<5 μg/dL) underscores methodological challenges, including potential reverse causality or unmeasured socioeconomic modifiers. Of interest is the particularly pronounced risk of ECC associated with lead exposure in primary teeth, where even slight increases in blood lead levels can elevate the risk of dental caries [79,85].

The biological mechanism linking lead exposure to ECC may be multifaceted. Lead accumulates in developing teeth and may disrupt enamel mineralization, though evidence remains equivocal. While hypomineralization could theoretically increase dental caries vulnerability, epidemiological data are inconsistent [86]. Like calcium, most absorbed lead accumulates in bone and teeth. Unlike bone, where lead can be released during remodeling processes, the accumulation of lead in the enamel is permanent and increases over time [10,61]. Furthermore, lead disrupts calcium metabolism, reduces salivary gland function [80], and may impair immune responses [87], all of which can exacerbate susceptibility to caries in children. Thus, lead’s primary contribution to ECC may lie in its disruption of salivary function or immunity rather than enamel structure [88,89,90].

Several critical measures have been implemented globally to reduce lead pollution and its harmful effects. One of the most significant actions was the worldwide phase-out of leaded gasoline, resulting in a substantial decline in airborne lead concentrations [91]. In parallel, many countries introduced regulations to limit lead levels in household paints [92] and the use of lead in toys, ceramics, cosmetics, and electronic products [93]. Blood lead level surveillance programs are now routinely conducted in numerous countries, targeting high-risk populations, to detect and address exposure at an early stage [94]. Efforts have also been made to remediate contaminated sites, such as those near smelters or battery recycling facilities, and to replace aging, lead-containing water pipes [95].

International organizations like the World Health Organization and the UNICEF have played a pivotal role in raising awareness about the dangers of lead exposure, especially in regions where informal battery recycling and the use of lead-glazed pottery are widespread [96]. Moreover, several donor-funded projects aim to reduce environmental lead contamination in low- and middle-income countries [97]. However, despite this global progress, significant challenges remain, particularly in low-income nations where law enforcement is weak and informal lead-related industries continue to thrive without adequate oversight [98]. High-income countries have, however, made significant strides in enforcing lead control policies, resulting in steep declines in population-level lead exposure and associated health risks [99], one of which may be ECC. In contrast, many low-income countries continue to struggle with limited regulatory capacity, weak enforcement mechanisms, and the persistence of informal lead-dependent industries that lack oversight [98]. Children in low-income countries are not only more likely to be exposed to lead from multiple sources, but are also less likely to benefit from surveillance, remediation, or preventive oral healthcare services [98].

## 5. Unsafe Water and Early Childhood Caries

Perfluoroalkyl acids (PFAAs) are a class of synthetic compounds widely used as surfactants, stain repellents, and firefighting foams in industrial products [100]. These chemicals contaminate drinking water supplies [101] and may be present in household dust [102]. Emerging evidence suggests that certain PFAAs, such as perfluorodecanoic acid, may interfere with normal enamel development, potentially increasing susceptibility to ECC [103].

While the exact biological mechanisms underpinning this association are not yet fully understood, several plausible pathways have been proposed. One possible route is through the disruption of dentine mineralization. PFAAs lower thyroid hormone levels [104,105], which play a crucial role in bone metabolism, a process closely related to dentine mineralization [106]. Inadequate thyroid hormone levels can impair tooth development, resulting in enamel hypoplasia [107], a significant risk factor for ECC [108]. In addition, some PFAAs exhibit immunotoxic properties, which may compromise the host’s immune response to cariogenic bacteria [109]. These compounds may also interfere with the hormonal regulation of salivary gland function [110], resulting in reduced salivary flow, dry mouth, and diminished oral clearance [111]. However, the current evidence base remains limited, while the association cannot be established in all cases [112].

ECC remains highly prevalent in low- and middle-income countries, where unsafe drinking water, environmental toxins, and inadequate sanitation are more prevalent and disproportionately affect children [113]. Marginalized populations, including rural communities, urban slums, and Indigenous groups, are more likely to live in environments with higher exposure to pollutants such as lead, mercury, and PFAAs, which have been increasingly implicated in adverse oral health outcomes. The inequitable distribution of environmental hazards reflects broader structural inequalities that disproportionately affect the most vulnerable, who are also more severely affected by ECC. Studies exploring the possible link between unsafe water and ECC are needed.

## 6. Ecovitality and Early Childhood Caries

Broader ecological degradation further exacerbates ECC risk by eroding “ecovitality,” the resilience of ecosystems that underpin clean water, air, and food security [114,115]. Environmental stressors (pollution, climate change, biodiversity loss) may amplify ECC by reducing access to clean water [16], thereby affecting oral hygiene, altering food systems, increasing reliance on processed foods high in sugars [116], and increasing exposure to endocrine disruptors such as Bisphenol A and heavy metals that may affect tooth development or saliva composition [117]. Moreover, air pollution (PM2.5) and ecosystem disruption compound these risks. Exposure to PM2.5 is associated with higher childhood caries rates [118], likely due to oxidative stress, inflammation, or enamel hypoplasia [119]. Degraded freshwater systems reduce access to optimally fluoridated water [120], although excessive fluoride from natural sources or unregulated programs can lead to dental fluorosis, which in severe cases increases caries risk. Soil depletion reduces the availability of nutrient-dense foods essential for enamel strength (e.g., calcium, vitamin D) [121,122]. Sadly, marginalized populations often face a “double burden” of environmental injustice (polluted neighbourhoods) [123] and limited access to dental care [124], amplifying ECC disparities. Extreme weather events can also disrupt healthcare access and exacerbate food insecurity, thereby indirectly increasing the risk of ECC [125].

Furthermore, pollutants such as PM2.5, ozone, and smog block Ultraviolet B radiation, limiting the skin’s ability to produce vitamin D, especially in densely populated and industrialized urban areas [126]. Agricultural burning of sugarcane fields releases PM2.5 and black carbon, contributing to regional air pollution [127] and possibly ECC. Geographic and seasonal factors further restrict sun exposure. Vitamin D is essential for enamel mineralization and immune regulation in early childhood. Deficiency during critical periods can impair enamel formation, leading to enamel hypoplasia and greater vulnerability to ECC [128]. It also weakens mucosal immunity and may disrupt salivary composition, reducing the mouth’s natural defenses against cariogenic bacteria like *Streptococcus mutans* [129]. Low vitamin D levels also directly increase the risk for ECC [130]. Vitamin D deficiency is geographically dependent [131]. The potential link between air pollution, reduced ultraviolet B radiation exposure, diminished vitamin D synthesis, and increased risk of ECC is a biologically plausible yet underexplored pathway at the crossroads of environmental health, nutrition, and oral epidemiology.

Similarly, air pollution has been independently associated with ECC through mechanisms like oxidative stress and compromised enamel integrity [132,133,134]. These suggest overlapping risk factors for ECC that merit deeper investigation. Disentangling these connections is, however, challenging due to confounders such as diet, socioeconomic status, healthcare access, and the threat to oral health by pollution through other pathways, such as through heavy metal toxicity [118]. Addressing these gaps could clarify mechanisms and inform effective public health responses.

## 7. Discussion

Environmental drivers of ECC, including lead toxicity, air pollution, and degraded water systems, disproportionally threaten children in marginalized communities, perpetuating cycles of ill health that contravene SDG 3.1’s mandate to end preventable child deaths. Addressing these determinants is inseparable from achieving SDG 3.1. Table 1 provides a summary of the possible pathways by which the discussed environmental factors may increase the risk for ECC.

Table 2 provides a summary of the level of evidence of the study findings. Tobacco smoke has a strong association with caries risk as shown in consistent meta-analyses, though residual confounding, like socioeconomic factors, may affect estimates. Lead exposure shows moderate evidence with dose-response links but inconsistent enamel defect data. PFAAs have low evidence, mainly from in vitro studies, lacking longitudinal human research. PM2.5 and vitamin D deficiency have moderate evidence supported by epidemiology and biological insights, but socioeconomic confounders limit conclusions. Overall, cautious interpretation is needed, and more longitudinal and mechanistic studies are crucial to clarify these associations that can guide effective policies.

Figure 1 is a conceptual framework providing a summary of the link between environmental factors and ECC, with a focus on vulnerable populations. It highlights that key upstream interactions, such as lead and PFA exposure, unsafe water, and ecological degradation, can disrupt biological functions and limit access to essential resources like fluoride and nutritious food. These exposures act through biological mechanisms (e.g., enamel defects, salivary dysfunction, immune impairment) and behavioral/social pathways (e.g., food insecurity, high sugar intake, reduced healthcare access). Moderating factors like socioeconomic inequities, environmental injustice, and weak infrastructure amplify these effects. Together, they lead to increased ECC prevalence, severity, and untreated cases, disproportionately affecting marginalized communities.

We postulate that improvements in global policies and programs that have led to reductions in household air pollution (including tobacco smoke and lead exposure), improvements in water safety, sanitation, hygiene practices, and reductions in child growth failure between 2010 and 2019 [7] may have contributed to a decline in the global prevalence of ECC. A 7.9% global reduction in the prevalence of untreated caries in primary teeth from 1990 to 2017 was reported, though factors associated with the observed decline are unknown [6]. The current evidence, however, remains limited and largely correlative, informed by observational studies that require validation through rigorous longitudinal cohort studies that can establish temporal sequence and causality. Furthermore, mechanistic research is essential to elucidate the precise biological pathways through which exposures like PFAAs or PM2.5 influence ECC pathogenesis. Future studies must also prioritize the development of standardized metrics for environmental exposures (e.g., ‘ecovitality’) and integrate them with oral health surveillance data to reduce measurement error and potential confounding. To date, no studies have empirically evaluated the impact of ecological restoration efforts, such as clean water initiatives, on ECC.

The interconnectedness between the environment and biological, behavioral, and social pathways for ECC implies that a lot more than environmental controls are needed to reduce the global prevalence of ECC. Unfortunately, countries with a low sociodemographic index are often the least equipped with the necessary resources to provide preventive and curative services for ECC [135]. These countries also tend to lack universal health coverage, a key enabler of equitable access to oral healthcare services and a mitigating factor for untreated ECC [136]. Global inequalities contribute to an uneven distribution of essential resources, such as skilled health personnel, financial assets, infrastructure, and access to goods and services, ultimately hampering efforts to control ECC [137].

In addition, middle-income countries undergoing economic transitions may face an elevated risk of ECC due to several intersecting factors. These include increased per capita sugar consumption [13], slower progress in adopting child-friendly and human rights–based health policies [138], and delayed investments in infrastructure that support both child access to oral healthcare [139] and environmentally sustainable systems [140]. Furthermore, efforts to reduce housing and income poverty may yield indirect benefits for ECC prevention and control by addressing some of the underlying social determinants of oral health [141].

Addressing the role of environmental toxins in ECC not only expands the scientific understanding of the disease’s aetiology. The idea that oral health risks are shaped by both individual behaviours and planetary health forces is gaining traction [142]. The planetary health perspective expands the lens of ECC research to include how global forces such as climate change, pollution, exposure to environmental toxins, urbanization, food systems, global sugar supply chains, and inequitable access to clean water and oral healthcare shape the environments in which children grow up [142] and influence their ECC risk. This shift acknowledges that children’s oral health is influenced by the health of the planet and the socio-environmental conditions in which their families live.

Addressing ECC also demands confronting the global sugar supply chain’s environmental externalities. Policies promoting sustainable agriculture (e.g., agroecology, reduced pesticide use) and equitable trade practices could mitigate habitat destruction and pollution, while simultaneously reducing sugar availability in at-risk communities. Such measures align with SDG 12 (Responsible Consumption) and SDG 15 (Life on Land), illustrating how oral health intersects with broader ecological sustainability [19,22,139].

This developed conceptual framework offers a comprehensive guide for investigating the complex, multifactorial predisposing factors for ECC by integrating environmental determinants alongside biological, behavioral, and structural determinants. It promotes the exploration of intermediate pathways using biomarker studies, longitudinal cohorts, and community surveys. By incorporating moderating factors such as socioeconomic inequities and policy failures into planned studies, the framework can facilitate studies that conduct multilevel modelling and policy analyses to uncover how systemic disadvantages heighten risk and limit protection. It highlights the importance of testing interactions, such as the combined effect of lead exposure and poor nutrition, using pathway analysis and structural equation modelling, and encourages interdisciplinary collaboration. In addition, conducting studies using community-based participatory research and stratified analyses can help to identify and support the most affected groups and address the issues of equity. The framework also promotes translational research, including cost-benefit analyses and evaluations of integrated policies that combine dental and environmental health strategies. Overall, it is a call for holistic, interdisciplinary, and equity-focused research to inform effective interventions and policy solutions that address the root causes of ECC disparities.

This evolving framework on ECC control also aligns with the One Health approach that underscores the complementarity of environmental, social, and biological perspectives, rejecting siloed explanations in favor of holistic solutions. It emphasizes the interconnectedness of human, animal, and environmental health [143,144]. For example, pesticide runoff from sugarcane fields contaminates water sources, affecting aquatic ecosystems and human microbiota, while habitat loss increases human-wildlife contact, potentially altering zoonotic disease patterns [145]. Traditionally used in the context of zoonotic diseases, antimicrobial resistance, and food safety, the One Health framework is increasingly being applied to non-communicable diseases [144] like ECC to reflect the complex web of determinants that shape health outcomes. The One Health approach encourages us to move beyond narrow biomedical models that focus only on individual behaviors and, instead, consider how broader ecological, societal, and environmental factors interact to influence oral health, including the risk for ECC. It encourages tackling the underlying determinants of health inequities—like poverty, food insecurity, and environmental injustice—while advancing sustainable, ecosystem-friendly interventions.

Sadly, the 2021 systematic analysis, revealing that age-standardized prevalence and disability-adjusted life years (DALYs) for untreated caries in primary teeth have remained unchanged over the last decade [146], raises important concerns about the overlooked role of environmental health determinants in shaping the global burden of oral disease. This stagnation in the global dental caries prevalence stands in stark contrast to global health efforts aimed at curbing modifiable risk factors for dental caries. The disconnect suggests three possibilities: current interventions are poorly targeted or insufficiently implemented; behavioural and structural determinants of oral health are dominant causes of ECC, with environmental toxins likely operating as effect modifiers rather than primary causes, interacting with sugar intake and fluoride access. The third possibility could be that the influence of environmental determinants is not being systematically measured, as environmental health indicators are often absent from global oral health datasets. This absence can obscure localized improvements or declines. Without comprehensive, interdisciplinary surveillance systems that connect environmental and oral health data, researchers risk missing key drivers of caries persistence, particularly in under-resourced regions where environmental hazards are most pronounced. Thus, while the findings confirm the continuing global burden of untreated dental caries, they simultaneously underscore the urgent need for research that explicitly prioritizes environmental health linkages using approaches that integrate environmental exposure science (such as exposomics) [147] and developing regionally sensitive data systems capable of capturing how ecological and policy-level changes that shape oral disease trajectories over time. Without closing these knowledge gaps, the transformative potential of environmental health interventions in reducing oral diseases such as caries will remain largely unrealized.

The 2021 systematic analysis also revealed that the African and Eastern Mediterranean World Health Organization regions experienced the most significant increases in prevalent cases and DALYs for untreated caries in primary teeth [146]. The current study’s hypotheses may align with these findings, as environmental health is notably worse in regions with the highest rise in untreated dental caries in primary teeth [148,149]. In addition, other political and economic factors, such as gross national income, inequality index, life expectancy [150], universal health coverage, and oral health expenditure [136], may also contribute to the higher prevalence of ECC in these regions.

Reducing exposure to environmental pollutants requires coordinated action across sectors, including public health, environmental regulation, water and sanitation, and urban planning, supported by policies that prioritize marginalized communities. Efforts to improve access to safe drinking water, regulate hazardous substances, and integrate oral health into broader environmental health initiatives will be critical to reducing ECC prevalence globally and narrowing the gap in oral health disparities. It involves not only scaling up technical interventions but also investing in health systems strengthening, policy reform, and community-led initiatives tailored to local contexts. Addressing ECC from a planetary health lens requires multilevel, interdisciplinary solutions that target not only individual behavior but also improve environmental sustainability, reduce inequality, and promote health-supportive policies globally. Until such equity is achieved, the burden of environment-induced ECC risk will continue to fall disproportionately on the world’s most vulnerable children.

## 8. Conclusions

While ECC prevention commonly focuses on proximal risk factors (direct and immediate risk contributors) such as sugar consumption, oral hygiene, and tooth resistance to demineralization, distal environmental determinants significantly shape individuals’ ability to manage these risks. Our synthesis does not diminish the importance of these proximal risk factors. Instead, it proposes environmental determinants as vital, interconnected, complementary modifiable drivers that exacerbate biological susceptibility, constrain access to protective resources, and may be a barrier to achieving SDG 3.1. By prioritizing ecosystem resilience (ecovitality) and One Health strategies, policymakers can concurrently advance oral health, child survival, and sustainable development. A primary imperative for future research is to move beyond correlation and quantify the population-attributable risk of environmental determinants for ECC. This requires well-designed longitudinal studies and natural experiments that assess the efficacy of real-world policies (e.g., lead pipe removal, air quality improvements) on ECC outcomes. Integrating environmental monitoring into oral health surveillance is essential to track equity-focused progress. Future interventions must also target the environmental drivers of sugar overconsumption, such as unsustainable agricultural practices, while advancing policies that prioritize ecosystem restoration and equitable food systems.

## Figures and Tables

**Figure 1 dentistry-13-00484-f001:**
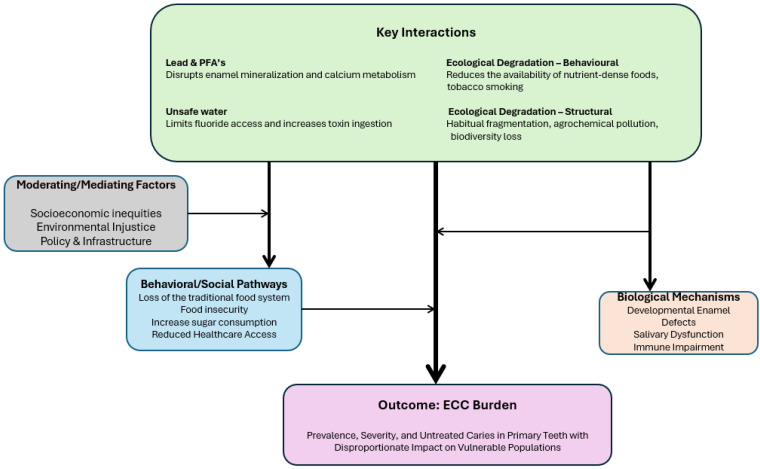
Conceptual Framework on the link between environmental determinants of health and ECC.

**Table 1 dentistry-13-00484-t001:** Environmental Determinants of ECC: Pathways and Evidence.

Exposure	Direct Biological Pathway	Indirect Social Pathway	Key Supporting Studies
Tobacco smoke	Enamel hypoplasia, decreased saliva	Maternal smoking → increased sugar use	[9,34]
Lead	Disrupted Ca^2+^ metabolism, increased biofilm virulence	Limited access to care in polluted areas	[80,86]
PFAAs	Thyroid disruption → enamel defects	Contaminated water → no fluoride	

**Table 2 dentistry-13-00484-t002:** Strength of Evidence for Key Environmental Exposures and Their Limitations.

Exposure	Evidence Strength	Key Supporting Studies	Major Limitations
Tobacco Smoke	High (consistent meta-analyses)	[9,28,38]	Residual confounding (e.g., socioeconomic status) may inflate risk estimates.
Lead	Moderate (dose-response with some mechanistic gaps)	[80,86]	Inconsistent enamel defect data; mechanistic uncertainty
PM2.5 Vitamin D	Moderate (epidemiological and biological evidence)	[121,128,129,130]	Confounding by socioeconomic status; difficulty isolating independent effects
PFAAs	Low (mechanistic plausibility, limited human data)	[104,110]	Scarcity of longitudinal human studies; reliance on in vitro evidence

## Data Availability

No new data were created or analyzed in this study. Data sharing is not applicable to this article.

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
