# Peer review of "Environmental Determinants of Early Childhood Caries: A Narrative Synthesis of Observational Evidence and Implications for Global Policy"

_dentistry, 2025, doi:10.3390/dj13110484_

Round 1
Reviewer 1 Report
Comments and Suggestions for Authors
The manuscript dentistry-3692204 has an interesting proposal to seek a relationship between environmental factors and dental caries. It is a “Perspective” article and therefore has a satisfactory level of freedom to present different paradigms and novel viewpoints. Despite this, the article fails in several points to consider:
1) As an article that aims to present global policy for ECC, I consider the absence of the Bangkok Declaration on ECC a serious flaw (Pitts, N, Baez, R, Diaz-Guallory, C, et al. Early Childhood Caries: IAPD Bangkok Declaration. Int J Paediatr Dent. 2019;29:384‐386; DOI: 10.1111/ipd.12490). This is a recent document, constructed and based on scientific evidence material, referenced by the WHO, and cannot be neglected. In addition, the document provides a lay and clinical definition of ECC that is of upmost importance for the whole article.
2) The authors consider sugar to be a “proximal risk factor” (line 395). I donot accept this definition/classification since sugar is the true etiological driving force for dental caries. Hence, I suggest considering sugar as a primary etiological factor fo dental caries assuming its role as indicated by several publications in this topic (please see the scientific definition for caries in the Bangkok document: “Dental caries is a biofilm‐mediated, sugar‐driven, multifactorial, dynamic disease…”
Moreover, I consider that the potential relation between sugar consumption and environmental factors is missing. For instance, there is substantial evidence that massive sugar cane cultivation has led to significant habitat fragmentation and biodiversity loss in many countries. As a result, air, water and soil degradation and loss of quality of life, and the potential risk for many chronic disease related to environmental factors.
I suggest that the authors adopt the concept of caries as a chronic noncommunicable disease as a bridge for integrating ECC global policies integrated with other chronic diseases in childhood (e.g. obesity). This is also a missing point in the article. To provide more evidence on these topics here you have some interesting papers to consider for inclusion in the manuscript:
Shepon, Alon, et al. "The environmental and social opportunities of reducing sugar intake." Proceedings of the National Academy of Sciences 121.48 (2024): e2314482121.
Gracner T, et al. Exposure to sugar rationing in the first 1000 days of life protected against chronic disease. Science. 2024;386(6725):1043-1048. doi: 10.1126/science.adn5421.
Abanto, Jenny, et al. "Impact of the first thousand days of life on dental caries through the life course: a transdisciplinary approach." Brazilian Oral Research 36 (2022): e113.
3) The authors assume that there is a clear direct association of ECC and a more fragile enamel. This is partially true since sugar and other oral factors can modulate the caries evolution particularly if there is an enamel breakdown.
Thus, I would be more cautious in this regard because topical etiological factors in the oral environment (oral hygiene, fluorides, sugar) are clearly superior in terms of biological plausibility for dental caries evolution. On the other hand, interference with saliva production and its quality, and loss of immunity are indeed acceptable etiological direct factors and do have strong potential to modulate the progression of caries. Finally, among the items discussed in the article, I would be more cautious on this issue and especially on the relationship between ECC, caries impact and lead (considering the formation of a more fragile tooth). After all, it may have a dose-effect issue with blood lead levels (<5 μg/dL)] (see Kim, YS., Ha, M., Kwon, HJ. et al. Association between Low blood lead levels and increased risk of dental caries in children: a cross-sectional study. BMC Oral Health 17, 42 (2017). https://doi.org/10.1186/s12903-017-0335-z), and a complex interplay of environmental and oral factors. The authors neglect that this relationship in the model presented in the article may not exist as proposed. To support the ms, here are few references to present the topic in a more impartial way.
No data was found that would give evidence of a relationship between lead concentration and enamel defects in either of the areas studied. No relationship was found between lead and dental caries in the industrial area, thus emphasizing that more studies of such relationships are needed.
Gomes, Viviane Elisângela, et al. "Lead level, enamel defects and dental caries in deciduous teeth." Revista de saude publica 38 (2004): 716-722.
This study shows a lack of association between exposure to lead between the ages of 1–4 years of age and dental caries in permanent dentition later in life. Other covariates, such as age and sugar consumption, appeared to play a more prominent role in caries development.
Yepes, Juan F., et al. "Blood levels of lead and dental caries in permanent teeth." Journal of public health dentistry 80.4 (2020): 297-303.
Therefore, the higher prevalence of DMFS in permanent tooth than caries in deciduous tooth is more influenced by other factors such as social and health-care conditions rather than causal risk factors such as lead exposure.
Kim, YS., Ha, M., Kwon, HJ. et al. Association between Low blood lead levels and increased risk of dental caries in children: a cross-sectional study. BMC Oral Health 17, 42 (2017). https://doi.org/10.1186/s12903-017-0335-z
4) Figure 1 must be revisited. I cannot understand that “lead exposure” and “unsafe water” are “Upstream environmental determinants” and “key interactions” simultaneously in this model. In addition, it is a matter of concern to classify “sugar” as an “intermediate pathway” and divide social determinants as pathway and moderation factors at the same time as well. As a reader, I would expect a better explanatory figure as fig 2 of a previous publications of the group (Crystal et al. BMC Oral Health (2024) 24:769 https://doi.org/10.1186/s12903-024-04535-9)
As a suggestion, it might be interesting to see other dental caries models of risk
Martignon S, Roncalli AG, Alvarez E, Aránguiz V, Feldens CA, Buzalaf MAR. Risk factors for dental caries in Latin American and Caribbean countries. Braz Oral Res. 2021 May 28;35(suppl 01):e053. doi: 10.1590/1807-3107bor-2021.vol35.0053
5) Finally, there is a clear limitation of the manuscript, that is: presenting the scientific evidence for the association of the environmental factors and ECC that were selected by the group. I noticed that many references are comprehensive, narrative, and scoping reviews. There are only three cohort studies. This is a clear indication that this scientific topic needs to be developed with other experimental/observational methods to increase scientific evidence.
Author Response
Reviewer 1
The manuscript dentistry-3692204 has an interesting proposal to seek a relationship between environmental factors and dental caries. It is a “Perspective” article and therefore has a satisfactory level of freedom to present different paradigms and novel viewpoints. Despite this, the article fails in several points to consider:
Response: We sincerely thank the reviewer for recognizing the novelty of our proposed framework linking environmental determinants to ECC. We agree that a “Perspective” article should balance conceptual innovation with empirical grounding. In response, we have revised the manuscript to enhance both its theoretical clarity and evidentiary support. We are grateful for the reviewer’s thoughtful and constructive suggestions, which have significantly improved the manuscript. Below, we provide a point-by-point response to each comment.
1) As an article that aims to present global policy for ECC, I consider the absence of the Bangkok Declaration on ECC a serious flaw (Pitts, N, Baez, R, Diaz-Guallory, C, et al. Early Childhood Caries: IAPD Bangkok Declaration. Int J Paediatr Dent. 2019;29:384‐386; DOI: 10.1111/ipd.12490). This is a recent document, constructed and based on scientific evidence material, referenced by the WHO, and cannot be neglected. In addition, the document provides a lay and clinical definition of ECC that is of upmost importance for the whole article.
Response: We sincerely thank the reviewer for this critical observation. We acknowledge the significant oversight in not referencing the 2019 Bangkok Declaration on Early Childhood Caries (ECC). We agree that this landmark document, endorsed by the International Association of Paediatric Dentistry (IAPD) and recognized by the WHO, provides essential standardized definitions and a comprehensive framework for global ECC policy. While our manuscript utilizes the foundational 1999 Drury workshop definition of ECC ("untreated decayed, filled, or missing primary teeth in children under 72 months"), we fully recognize that the Bangkok Declaration refines and expands this definition to enhance clinical utility and global applicability. To address this gap and strengthen the global policy context, we revised paragraph 1. Critically, the Bangkok Declaration does not contradict the Drury definition but builds upon it, maintaining the core elements (age <6, primary teeth, caries experience). Our manuscript’s environmental focus (e.g., lead, pollution, water safety) directly operationalizes the Bangkok Declaration’s mandate to address upstream inequities. Thus, integrating this document strengthens—rather than conflicts with—our synthesis.
2) The authors consider sugar to be a “proximal risk factor” (line 395). I do not accept this definition/classification since sugar is the true etiological driving force for dental caries. Hence, I suggest considering sugar as a primary etiological factor for dental caries, assuming its role as indicated by several publications in this topic (please see the scientific definition for caries in the Bangkok document: “Dental caries is a biofilm‐mediated, sugar‐driven, multifactorial, dynamic disease…” Moreover, I consider that the potential relationship between sugar consumption and environmental factors is missing. For instance, there is substantial evidence that massive sugar cane cultivation has led to significant habitat fragmentation and biodiversity loss in many countries. As a result, air, water, and soil degradation and loss of quality of life, and the potential risk for many chronic diseases related to environmental factors.
I suggest that the authors adopt the concept of caries as a chronic noncommunicable disease as a bridge for integrating ECC global policies, integrated with other chronic diseases in childhood (e.g. obesity). This is also a missing point in the article. To provide more evidence on these topics here you have some interesting papers to consider for inclusion in the manuscript:
Shepon, Alon, et al. "The environmental and social opportunities of reducing sugar intake." Proceedings of the National Academy of Sciences 121.48 (2024): e2314482121.
Gracner T, et al. Exposure to sugar rationing in the first 1000 days of life protected against chronic disease. Science. 2024;386(6725):1043-1048. doi: 10.1126/science.adn5421.
Abanto, Jenny, et al. "Impact of the first thousand days of life on dental caries through the life course: a transdisciplinary approach." Brazilian Oral Research 36 (2022): e113.
Response: We sincerely thank the reviewer for their insightful comments, which significantly strengthen our manuscript's conceptual framework and global policy implications. First, to note that when we say "sugar is a proximal risk factor," we mean that sugar consumption is a direct and immediate contributor to a health problem, especially in the short to medium term. We think that the use of the concept ‘proximal factor’ is correct in the current context. We fully agree with the other points raised and have revised multiple sections of the manuscript accordingly.
3) The authors assume that there is a clear direct association between ECC and a more fragile enamel. This is partially true since sugar and other oral factors can modulate the caries evolution, particularly if there is an enamel breakdown. Thus, I would be more cautious in this regard because topical etiological factors in the oral environment (oral hygiene, fluorides, sugar) are clearly superior in terms of biological plausibility for dental caries evolution. On the other hand, interference with saliva production and its quality, and loss of immunity are indeed acceptable etiological direct factors and do have strong potential to modulate the progression of caries. Finally, among the items discussed in the article, I would be more cautious on this issue, and especially on the relationship between ECC, caries impact, and lead (considering the formation of a more fragile tooth). After all, it may have a dose-effect issue with blood lead levels (<5 μg/dL)] (see Kim, YS., Ha, M., Kwon, HJ. et al. Association between Low blood lead levels and increased risk of dental caries in children: a cross-sectional study. BMC Oral Health 17, 42 (2017). https://doi.org/10.1186/s12903-017-0335-z), and a complex interplay of environmental and oral factors. The authors neglect that this relationship in the model presented in the article may not exist as proposed. To support the ms, here are a few references to present the topic in a more impartial way.
No data was found that would give evidence of a relationship between lead concentration and enamel defects in either of the areas studied. No relationship was found between lead and dental caries in the industrial area, thus emphasizing that more studies of such relationships are needed.
Gomes, Viviane Elisângela, et al. "Lead level, enamel defects and dental caries in deciduous teeth." Revista de saude publica 38 (2004): 716-722.
This study shows a lack of association between exposure to lead between the ages of 1–4 years of age and dental caries in permanent dentition later in life. Other covariates, such as age and sugar consumption, appeared to play a more prominent role in caries development.
Yepes, Juan F., et al. "Blood levels of lead and dental caries in permanent teeth." Journal of public health dentistry 80.4 (2020): 297-303.
Therefore, the higher prevalence of DMFS in permanent tooth than caries in deciduous tooth is more influenced by other factors such as social and health-care conditions rather than causal risk factors such as lead exposure.
Kim, YS., Ha, M., Kwon, HJ. et al. Association between Low blood lead levels and increased risk of dental caries in children: a cross-sectional study. BMC Oral Health 17, 42 (2017). https://doi.org/10.1186/s12903-017-0335-z
Response: We thank the reviewer for drawing attention to the complex relationship between environmental toxins, enamel integrity, and ECC. In response, we have revised the manuscript to present enamel fragility as a contributing factor rather than a primary cause, giving greater weight to more established topical etiological factors such as sugar exposure and fluoride availability. We have also addressed the conflicting evidence on lead exposure by incorporating studies with differing findings, highlighting the dose-dependent nature of these effects. We observed, however, that there seems to be consistent evidence on the association between lead and dental caries in the primary dentition, while less so in the permanent dentition. In addition, we have updated the conceptual framework (Figure 1) to illustrate the interactions between lead exposure, behavioural pathways, and relevant confounders. These revisions aim to present a more balanced and evidence-informed discussion, while maintaining the central argument that environmental determinants influence ECC risk within a broader socioecological context
4) Figure 1 must be revisited. I cannot understand that “lead exposure” and “unsafe water” are “Upstream environmental determinants” and “key interactions” simultaneously in this model. In addition, it is a matter of concern to classify “sugar” as an “intermediate pathway” and divide social determinants as pathway and moderation factors at the same time as well. As a reader, I would expect a better explanatory figure as fig 2 of a previous publications of the group (Crystal et al. BMC Oral Health (2024) 24:769 https://doi.org/10.1186/s12903-024-04535-9)
As a suggestion, it might be interesting to see other dental caries models of risk
Martignon S, Roncalli AG, Alvarez E, Aránguiz V, Feldens CA, Buzalaf MAR. Risk factors for dental caries in Latin American and Caribbean countries. Braz Oral Res. 2021 May 28;35(suppl 01):e053. doi: 10.1590/1807-3107bor-2021.vol35.0053
Response: Thank you for this thoughtful and constructive feedback. We appreciate your concerns regarding the clarity and consistency of Figure 1. In response, we have revised the figure to improve its conceptual coherence and visual clarity. The term "Upstream environmental determinants" has been retained to align with the manuscript’s framing, which draws on socioecological and One Health models. However, we have now restructured the figure to distinguish more clearly between environmental determinants (e.g., lead exposure, unsafe water) and the intermediate pathways through which they exert their influence, namely, biological and behavioral/social mechanisms. We acknowledge the confusion caused by classifying elements such as sugar consumption and social determinants across multiple categories. In the revised figure, we have refined these classifications. Sugar is now presented within the behavioural pathway, clearly linked to broader environmental and systemic factors. Similarly, socioeconomic conditions are now consistently framed as moderating/mediating factors that influence both exposure and vulnerability, rather than being duplicated across categories. We also reviewed the suggested literature, including the framework from Crystal et al. (2024) and Martignon et al. (2021), and drew inspiration from their clarity and layered structure. While our model differs in its One Health orientation and emphasis on upstream environmental drivers, we agree that these references offer valuable examples of risk models for dental caries and have used them to further refine our approach. We hope the revised figure provides a more coherent and explanatory representation of the framework underpinning our argument. Thank you again for the suggestion—it has strengthened the clarity and rigor of the manuscript.
Bottom of Form
5) Finally, there is a clear limitation of the manuscript, that is: presenting the scientific evidence for the association of the environmental factors and ECC that were selected by the group. I noticed that many references are comprehensive, narrative, and scoping reviews. There are only three cohort studies. This is a clear indication that this scientific topic needs to be developed with other experimental/observational methods to increase scientific evidence.
Response: We thank the reviewer for highlighting this important limitation. We fully acknowledge that the current evidence base linking environmental factors to ECC is limited, with only a few cohort studies. This reflects the emerging nature of the field, where direct empirical studies, particularly longitudinal and mechanistic research, remain scarce. Advancing the science requires a more robust body of primary research. In response, we have strengthened the manuscript’s Discussion and Conclusion sections to more clearly articulate this gap and called for interdisciplinary longitudinal studies, exposomics, and integrated cohort designs. Our revised conceptual framework (Figure 1) is intended to guide future research by mapping plausible biological and behavioral pathways linking environmental exposures to ECC. We did not write an exclusive study limitation section as may otherwise be expected.
Reviewer 2 Report
Comments and Suggestions for Authors
General Comments:
The manuscript addresses an important and timely topic—the role of environmental factors in early childhood caries (ECC). The introduction provides a comprehensive overview, linking environmental exposures such as tobacco smoke, lead, and pollution with ECC risk. The paper rightly emphasizes the need to consider environmental determinants beyond traditional behavioral and biological factors. The interdisciplinary approach and the introduction of concepts like ecovitality and One Health are promising for advancing research and public health policies.
However, several areas require clarification, expansion, or refinement to improve the manuscript's clarity, coherence, and scientific rigor. Below are detailed comments.
1. Definition and scope of ECC
-
The definition is clear and appropriately cited. However, the statement “The definition of ECC as a clinical entity helped to streamline the varied features…” could be supported by explaining briefly how the definition has impacted epidemiological or clinical research.
-
Suggestion: Add a sentence clarifying the practical implications of standardizing ECC definition for research and intervention design.
2. The introduction mentions reductions in behavioral risk factors over time but increases or stagnation in environmental risks. The sources [4], [5] are cited, but it would strengthen the argument to briefly state what kind of evidence supports these trends—are they from global data, specific regions, or population-based surveys?
3. Section 2: Smoked Tobacco and Early Childhood Caries
-
The meta-analyses and odds ratios cited support the association well. However, the conflicting evidence (studies showing no association) is only briefly mentioned and should be discussed more critically to assess possible reasons for discrepancies (e.g., study design, population differences).
-
Suggestion: Provide a more balanced discussion on conflicting findings and the quality of evidence.
4. Biological mechanisms
-
The mechanistic explanations are well presented, but references should be updated to the most recent reviews if available.
-
5. Lead Poisoning and Early Childhood Caries-Mechanisms and evidence
-
The biological explanations are solid. Consider highlighting which studies provide the strongest epidemiological evidence linking lead and ECC, including study designs and limitations.
-
Suggestion: Clarify the strength of the evidence base (e.g., longitudinal vs. cross-sectional).
-
6. Global interventions
-
The description of policies to reduce lead exposure is thorough. However, it would benefit from a more critical assessment of their effectiveness related to oral health outcomes specifically.
-
Suggestion: If data on lead reduction leading to ECC improvement exist, cite them.
- 7. Clarifying evidence strength and limitations,
- 8. Strengthening discussion on conflicting findings,
-
9. Better linking policy implications directly to ECC outcomes,
-
10. Emphasizing research gaps and future directions.
-
-
-
-
Author Response
Reviewer 2
General Comments:
The manuscript addresses an important and timely topic—the role of environmental factors in early childhood caries (ECC). The introduction provides a comprehensive overview, linking environmental exposures such as tobacco smoke, lead, and pollution with ECC risk. The paper rightly emphasizes the need to consider environmental determinants beyond traditional behavioral and biological factors. The interdisciplinary approach and the introduction of concepts like ecovitality and One Health are promising for advancing research and public health policies. However, several areas require clarification, expansion, or refinement to improve the manuscript's clarity, coherence, and scientific rigor. Below are detailed comments.
Response: We sincerely thank the reviewer for their thoughtful and encouraging comments.
We have carefully revised the manuscript to enhance its clarity, coherence, and scientific rigor. Below, we provide a point-by-point response to each comment.
- Definition and scope of ECC
- The definition is clear and appropriately cited. However, the statement “The definition of ECC as a clinical entity helped to streamline the varied features…”could be supported by explaining briefly how the definition has impacted epidemiological or clinical research.
- Suggestion: Add a sentence clarifying the practical implications of standardizing ECC definition for research and intervention design.
Response: Thank you for highlighting this opportunity for clarification. We have added a sentence to explain the practical implications of standardizing the ECC definition, emphasizing its role in enabling consistent epidemiological surveillance, cross-population comparisons, and targeted public health interventions. This revision strengthens the narrative by linking definitional clarity to real-world research and policy applications.
- The introduction mentions reductions in behavioral risk factors over time but increases or stagnation in environmental risks.The sources [4], [5] are cited, but it would strengthen the argument to briefly state what kind of evidence supports these trends—are they from global data, specific regions, or population-based surveys?
Response: Thank you for the suggestion. We revised the section to clarify that the trends cited are based on global data from large-scale burden of disease analyses, which synthesize evidence from diverse sources, including population-based surveys across 204 countries. This strengthens the global relevance of the observed shifts in behavioral and environmental risk factors.
- Section 2: Smoked Tobacco and Early Childhood Caries
- The meta-analyses and odds ratios cited support the association well. However, the conflicting evidence (studies showing no association) is only briefly mentioned and should be discussed more critically to assess possible reasons for discrepancies (e.g., study design, population differences).
- Suggestion: Provide a more balanced discussion on conflicting findings and the quality of evidence.
Response: We appreciate the reviewer’s constructive feedback. We have revised Section 3 to critically address conflicting evidence and strengthen the discussion of methodological limitations. A new paragraph discussing the conflicting evidence has been included in the section.
- Biological mechanisms
- The mechanistic explanations are well presented, but references should be updated to the most recent reviews if available.
Response: Thank you for the suggestion. We reviewed the literature and found the current references remain appropriate and relevant for supporting the described biological mechanisms
- Lead Poisoning and Early Childhood Caries-Mechanisms and evidence
- The biological explanations are solid. Consider highlighting which studies provide the strongest epidemiological evidence linking lead and ECC, including study designs and limitations. Suggestion: Clarify the strength of the evidence base (e.g., longitudinal vs. cross-sectional).
Response: Thank you for the suggestion. We have revised the opening paragraph of the section to highlight the strongest epidemiological evidence linking lead exposure to ECC, including study designs and associated limitations.
- Global interventions
- The description of policies to reduce lead exposure is thorough. However, it would benefit from a more critical assessment of their effectiveness related to oral health outcomes specifically. Suggestion: If data on lead reduction leading to ECC improvement exist, cite them.
Response: Thank you for the suggestion. Currently, there is no direct evidence linking lead reduction policies to improvements in ECC outcomes. The manuscript highlights this gap and discusses the underlying reasons for the limited evidence, emphasizing the need for further research in this area.
- Clarifying evidence strength and limitations. Strengthening discussion on conflicting findings. Better linking policy implications directly to ECC outcomes. Emphasizing research gaps and future directions.
Response: Thank you for these valuable suggestions. We have thoroughly revised the manuscript to strengthen the discussion on conflicting findings, clarify the evidence strength and limitations, better connect policy implications to ECC outcomes, and emphasize research gaps and future directions. We have included Table 1, which provides a summary of the strength of evidence on the associations between environmental risk factors and ECC.
Reviewer 3 Report
Comments and Suggestions for Authors
The article takes a look at environmental factors to explain dental caries, which can generate a very accurate discussion among scientific peers on the subject. The discussion needs to lead the reader to the real objective and understand that this environmental approach is complementary and is being proposed for debate among peers.
There are no ethical infringements.
The writing is logical and well-structured, following the order of execution of a narrative review.
The reading incites a lot of discussion, even about the basic concepts of dental caries, as it presupposes environmental factors as causal factors that have never been mentioned before. Perhaps the authors need to make their objective clearer and discuss the approaches presented to justify the need for this type of study of dental caries.
I believe that if we reformulate the discussion by focusing on these environmental factors within the causes of dental caries in a way that presents and discusses them rather than affirming them, the article will have value since it will encourage a discussion, strengthening the points presented.
Author Response
Reviewer 3
The article takes a look at environmental factors to explain dental caries, which can generate a very accurate discussion among scientific peers on the subject. The discussion needs to lead the reader to the real objective and understand that this environmental approach is complementary and is being proposed for debate among peers.
Response: We sincerely thank the reviewers for their insightful feedback. We agree that positioning the environmental perspective as a complementary framework rather than a standalone explanation is crucial for accurately contextualizing our findings within the broader ECC discourse. Revisions have been made to various sections of the manuscripts to clarify this intent.
There are no ethical infringements. The writing is logical and well-structured, following the order of execution of a narrative review.
Response: Thanks for the constructive feedback.
The reading incites a lot of discussion, even about the basic concepts of dental caries, as it presupposes environmental factors as causal factors that have never been mentioned before. Perhaps the authors need to make their objective clearer and discuss the approaches presented to justify the need for this type of study of dental caries.
Response: We sincerely thank the reviewer for their insightful feedback. We acknowledge that positioning environmental determinants as causal factors for ECC represents a paradigm shift from traditional caries aetiology models. We have clarified our objective and justified our approach.
I believe that if we reformulate the discussion by focusing on these environmental factors within the causes of dental caries in a way that presents and discusses them rather than affirming them, the article will have value since it will encourage a discussion, strengthening the points presented.
Response: We fully agree with this approach. The discussion has been extensively revised to present the environmental factors as suggestive rather than definitive contributors to ECC. The updated language encourages critical reflection, and the inclusion of the new Table 1 further supports this evidence-based, exploratory framing.
Reviewer 4 Report
Comments and Suggestions for Authors
Thank you for the opportunity to consider this interesting work.
The manuscript submitted for potential publication in Dentistry Journal aimed to analyze some factors connected with early childhood caries (ECC). But there is no clear aim.
Dear Authors,
The presented manuscript needs many additional revisions.
The factors you have chosen do not directly affect the level of ECC. Justify why these factors are important, provide percentages defining the size of the studied populations, the mechanism of influence on the development of ECC.
It seems that you connect all aspects of environmental pollution with caries.
You didn't prove this thesis in such a written paper.
Define the aim.
The manuscript needs severe editing of the content.
Major points
Abstract
There is no aim, so we do not know what your goal is for the paper.
Abstract is very long and you mention many aspects, from vitamins to radiation.
My advice is to define the aim, limit the description to some aspects.
Introduction
You introduce the disease, but the definition of ECC is not in agreement with AAPD and handbooks on the topic from many other scientific associations. It is interesting because you try to convince the reader that the paper is for use in global policy.
My advice is to define and describe how to understand globally the disease, and what the aim of the paper is.
„For children 0-9 years old, behavioural risk factors such as nutrition-related factors, including wasting underweight, and stunting, known to be associated with ECC, were reduced between 1900 and 2019 [4]. By contrast, environmental factors such as unsafe sanitation, handwashing, and ambient particulate matter increased, whereas household air pollution and unsafe water remained at the same level [5]. These environmental factors have implications for ECC. These factors are known risk factors for ECC „
Please provide the citations for those factors as the risk factors of ECC. Your thesis seems to be risky. Secondly, I do not think that suddenly writing about children up to 9 years is on topic for your paper.
The last part of the Introduction is not clear. What problem do you really want to introduce?
Smoked Tobacco and Early Childhood Caries
Please explain the mechanism of tobacco smoking and the risk of ECC. There is no tobacco smoking in clinical protocols for ECC.
You write that there is a significant moderate association between passive tobacco exposure and caries in children.
There are major risk factors for ECC. I understand that your goal is to focus on all potentially harmful factors.
So please, give a rationale for choosing those factors, as they are not directly responsible for ECC.
Then precise if you focus on the caries level, level of ECC. Presentation of the problem must be as precise as possible.
Lead Poisoning and Early Childhood Caries
„This is because lead interferes with enamel formation during tooth development, resulting in hypomineralized and structurally weaker enamel that is more vulnerable to caries [69, 70]. The prevalence of caries was higher in children whose primary teeth had lead than in those who did not.”
You focus here on enamel formation. I think that the whole paper is about enamel formation.
What was the prevalence of caries in this researcher's study?
Unsafe water and Early Childhood Caries
You mention that ECC remains highly prevalent in low- and middle-income countries, where unsafe drinking water, environmental toxins, and inadequate sanitation are more prevalent. Yes, but does it allow us to conclude that unsafe water is responsible for the level of ECC?
The Environment and Early Childhood Caries
You again discuss the water problem, maybe it would be good to connect this aspect to the previous paragraph?
You mention that epidemiological evidence links low vitamin D levels with increased ECC risk. I think that this aspect is not raised nowadays.
Discussion
You mention that no studies have empirically evaluated the impact of ecological restoration efforts, such as clean water initiatives, on ECC. And I suppose they would not, as caries is defined and risk factors are defined.
You miss citations in the chapter.Please add them.
It seems to be a polemic rather than a scientific paper.
Conclusions
You introduce Sustainable Development Goal 3.1 at the end of the paper. It must be explained at the beginning and could be the basis for the whole concept of the paper.
Author Response
Reviewer 4
Thank you for the opportunity to consider this interesting work. The manuscript submitted for potential publication in the Dentistry Journal aimed to analyze some factors connected with early childhood caries (ECC). But there is no clear aim. The presented manuscript needs many additional revisions.
Response: Thank you for highlighting this important point. We have now clearly stated the aim of the study in the revised manuscript: While ECC etiology has traditionally focused on behavioral and biological factors, this review examines emerging evidence on environmental determinants as modifiable risk amplifiers in vulnerable populations. Our objective is to synthesize observational data on these understudied pathways and their relevance to global policy. We argue that understanding these interactions is crucial for developing holistic interventions and fostering critical debate on the planetary health dimensions of oral disease.
The factors you have chosen do not directly affect the level of ECC. Justify why these factors are important, provide percentages defining the size of the studied populations, the mechanism of influence on the development of ECC.
Response: Thank you for raising this point. As outlined in the objective and discussed throughout the manuscript, the aim is to explore potential links between these environmental factors and ECC. While the direct impact may not yet be fully established, highlighting these possible associations is intended to stimulate further research and broaden understanding of factors that may significantly contribute to caries prevention
It seems that you connect all aspects of environmental pollution with caries. You didn't prove this thesis in such a written paper.
Response: Thank you for the feedback. This narrative review explores possible associations between environmental factors and ECC, not to claim causality but to propose a multifactorial framework. We synthesize observational and mechanistic evidence showing how exposures like tobacco smoke, lead, PFAAs, and PM2.5 may increase ECC risk, especially in vulnerable populations, by disrupting tooth development, immune function, and access to protective resources. The revised manuscript strengthens specificity, clarifies intent, and highlights key research gaps.
Define the aim.
Response: Our objective is to synthesize observational data on the understudied environmental pathways and their implications for global policy.
The manuscript needs severe editing of the content.
Response: Thank you for the feedback. We have dedicated significant effort to revising the manuscript in response to the reviewers' comments. We appreciate the thorough review, which has greatly contributed to strengthening the paper. Below is our point-by-point response to the reviewer’s comments.
Major points
Abstract
There is no aim, so we do not know what your goal is for the paper. Abstract is very long and you mention many aspects, from vitamins to radiation. My advice is to define the aim, limit the description to some aspects.
Response: Thank you for the helpful feedback. We have now clearly stated the aim of the study and revised the abstract to make it more focused and concise.
Introduction
You introduce the disease, but the definition of ECC is not in agreement with AAPD and handbooks on the topic from many other scientific associations. It is interesting because you try to convince the reader that the paper is for use in global policy. My advice is to define and describe how to understand globally the disease, and what the aim of the paper is.
Response: Thank you for the observation. We have now included reference to the Bangkok Declaration to support a globally relevant understanding of ECC and better align the definition with international guidance.
„For children 0-9 years old, behavioural risk factors such as nutrition-related factors, including wasting underweight, and stunting, known to be associated with ECC, were reduced between 1900 and 2019 [4]. By contrast, environmental factors such as unsafe sanitation, handwashing, and ambient particulate matter increased, whereas household air pollution and unsafe water remained at the same level [5]. These environmental factors have implications for ECC. These factors are known risk factors for ECC „ Please provide the citations for those factors as the risk factors of ECC. Your thesis seems to be risky. Secondly, I do not think that suddenly writing about children up to 9 years is on topic for your paper.
Response: Thank you for this important feedback. We have also added appropriate citations [6, 7] to support the link between the mentioned environmental factors and ECC, and reworded the statement to reflect the exploratory nature of the associations rather than definitive claims. The reference to age 9 years is for primary teeth, which have some relationship with the study of ECC.
The last part of the Introduction is not clear. What problem do you really want to introduce?
Response: Thank you for the comment. We have revised and clarified the final part of the Introduction to better articulate the core problem the paper addresses.
Smoked Tobacco and Early Childhood Caries
Please explain the mechanism of tobacco smoking and the risk of ECC. There is no tobacco smoking in clinical protocols for ECC. You write that there is a significant moderate association between passive tobacco exposure and caries in children. There are major risk factors for ECC. I understand that your goal is to focus on all potentially harmful factors.
So please, give a rationale for choosing those factors, as they are not directly responsible for ECC. Then precise if you focus on the caries level, level of ECC. Presentation of the problem must be as precise as possible.
Response: We appreciate the reviewer’s call for greater mechanistic clarity. Tobacco smoke raises ECC risk through enamel development disruption, impaired salivary function, and microbiome dysbiosis. These biological effects lower resistance to acid, reduce protective saliva, and increase cariogenic bacteria like Streptococcus mutans. While sugar and hygiene remain primary drivers, environmental toxins like tobacco act as effect modifiers that intensify susceptibility. We have revised the section to reflect these mechanisms and emphasize tobacco as a modifiable environmental risk.
Lead Poisoning and Early Childhood Caries
„This is because lead interferes with enamel formation during tooth development, resulting in hypomineralized and structurally weaker enamel that is more vulnerable to caries [69, 70]. The prevalence of caries was higher in children whose primary teeth had lead than in those who did not.” You focus here on enamel formation. I think that the whole paper is about enamel formation. What was the prevalence of caries in this researcher's study?
Response: 13.5% of dental caries occurring among children exposed to high lead levels and 9.6% of dental caries among children exposed to moderate lead levels, can be attributed to lead exposure
Unsafe water and Early Childhood Caries
You mention that ECC remains highly prevalent in low- and middle-income countries, where unsafe drinking water, environmental toxins, and inadequate sanitation are more prevalent. Yes, but does it allow us to conclude that unsafe water is responsible for the level of ECC?
Response: Thank you for the thoughtful comment. The paper does not assert a definitive conclusion that unsafe water causes ECC. Rather, it raises critical questions about how environmental exposures—such as unsafe water—may interact with biological and social factors to influence ECC risk. The aim is to prompt further investigation into these underexplored pathways, not to draw causal inferences.
The Environment and Early Childhood Caries
You again discuss the water problem, maybe it would be good to connect this aspect to the previous paragraph? You mention that epidemiological evidence links low vitamin D levels with increased ECC risk. I think that this aspect is not raised nowadays.
Response: Thank you for highlighting this. We agree that explicitly linking the discussion of water safety to the broader ecological context will improve cohesion. We will add a transitional sentence to clarify how unsafe water and PFAA exposure exemplify the degradation of "ecovitality" (e.g., compromised water systems) and amplify ECC risk through interconnected pathways. While vitamin D’s role in ECC is well-established biologically (enamel mineralization, immune function), its connection to environmental factors like air pollution (via reduced UVB exposure) is an emerging and actively researched pathway. We have clarified the contemporary relevance of this mechanism
Discussion
You mention that no studies have empirically evaluated the impact of ecological restoration efforts, such as clean water initiatives, on ECC. And I suppose they would not, as caries is defined and risk factors are defined. You miss citations in the chapter. Please add them. It seems to be a polemic rather than a scientific paper.
Response: Thank you for your feedback. We appreciate the concern regarding citations. However, as noted, there is a lack of empirical studies directly linking ecological restoration efforts—such as clean water initiatives—to ECC outcomes. In the absence of such literature, we aim to highlight this gap and encourage further inquiry. If specific references are required, we would welcome suggestions, but our intent is not to present a polemic, but to raise scientifically grounded questions where evidence is currently limited.
Conclusions
You introduce Sustainable Development Goal 3.1 at the end of the paper. It must be explained at the beginning and could be the basis for the whole concept of the paper.
Response: We appreciate this insightful feedback. The reviewer rightly emphasizes the strategic importance of Sustainable Development Goal (SDG) 3.1 as a foundational framework for this review. We agree that foregrounding SDG 3.1 strengthens the paper’s narrative by aligning environmental determinants of ECC with a globally recognized health equity imperative.
Reviewer 5 Report
Comments and Suggestions for Authors
This review examined the interplay between exposure to environmental toxins (e.g. lead, air pollution, perfluoroalkyl substances (PFAS), tobacco smoke, and unsafe water) and early childhood caries (ECC).
The structure of the abstract is acceptable; it clearly states the purpose of the manuscript.
Strengths:
The introduction provides an adequate justification for the importance of the topic and the necessity of this review. The statements of the introduction are supported by appropriate references.
Quality: The article is detailed and written appropriately. The English is easy to understand.
References: The review article is based on a large number of relevant publications that are consistent with the aim and content of the manuscript.
Novelty, significance: Although the topic is not new, this is precisely why it is important. It summarises a problem that is familiar to every dentist and highlights the fact that the real impact of environmental factors on oral health is still not receiving enough attention in public health, despite being a well-known issue.
Discussion: The discussion is well structured and highlights, establishes, and supports the importance of investigating the impact of environmental factors in future research on ECC, in addition to the well-known primary and secondary etiological factors.
The authors also emphasize the significance and importance of interdisciplinary studies.
The conclusion is based on the data summarised previously. The authors reasonably recommend examining the contribution of distal environmental factors to ECC risk for global policy, and suggest the possibility of a 'justice-oriented' approach to ECC control to identify vulnerable populations.
Weakness and my suggestions:
- It would be important to emphasize not only the lack of fluoride, but also the danger of overdose. (line 222)
Reviewer Overall Recommendations: The article is acceptable, after minor corrections.
Author Response
Reviewer 5
This review examined the interplay between exposure to environmental toxins (e.g. lead, air pollution, perfluoroalkyl substances (PFAS), tobacco smoke, and unsafe water) and early childhood caries (ECC). The structure of the abstract is acceptable; it clearly states the purpose of the manuscript.
Response: Thank you for your positive feedback on the abstract. We're glad to hear that the structure and clarity effectively conveyed the purpose of the manuscript. The revisions made have further strengthened the manuscript.
Strengths:
The introduction provides an adequate justification for the importance of the topic and the necessity of this review. The statements of the introduction are supported by appropriate references.
Quality: The article is detailed and written appropriately. The English is easy to understand.
Response: Thank you for your positive feedback. We’re pleased to hear that the introduction effectively conveyed the importance of the topic and that the overall quality, clarity, and language of the manuscript met your expectations.
References: The review article is based on a large number of relevant publications that are consistent with the aim and content of the manuscript.
Response: Thank you. We appreciate the acknowledgment and have ensured that the referenced publications align closely with the manuscript’s objectives and scope.
Novelty, significance: Although the topic is not new, this is precisely why it is important. It summarises a problem that is familiar to every dentist and highlights the fact that the real impact of environmental factors on oral health is still not receiving enough attention in public health, despite being a well-known issue.
Response: Thank you for this insightful comment. We agree that the familiar nature of the topic underscores its significance. We aimed to synthesize existing evidence to draw renewed attention to the underappreciated impact of environmental factors on oral health and to advocate for a stronger public health focus in this area.
Discussion: The discussion is well structured and highlights, establishes, and supports the importance of investigating the impact of environmental factors in future research on ECC, in addition to the well-known primary and secondary etiological factors. The authors also emphasize the significance and importance of interdisciplinary studies.
Response: Thank you for your positive feedback. We are glad the discussion effectively conveyed the need to expand research on ECC to include environmental influences and highlighted the value of interdisciplinary collaboration in addressing this complex public health issue. This has been further strengthened.
The conclusion is based on the data summarised previously. The authors reasonably recommend examining the contribution of distal environmental factors to ECC risk for global policy, and suggest the possibility of a 'justice-oriented' approach to ECC control to identify vulnerable populations.
Response: Thank you. We appreciate your recognition of the rationale behind our conclusion. We intended to emphasize the need for equity-driven strategies that address the broader environmental determinants of ECC, particularly for protecting vulnerable populations and informing global oral health policy.
Weakness and my suggestions:
It would be important to emphasize not only the lack of fluoride, but also the danger of overdose. (line 222)
Response: We sincerely thank the reviewer for this critical insight. The potential risks of fluoride overdose, particularly dental fluorosis, are indeed a significant public health concern that complements our discussion on fluoride deficiency. We agree that a balanced perspective on fluoride is essential for evidence-based policy and have included a sentence to this effect.
Reviewer Overall Recommendations: The article is acceptable, after minor corrections.
Response: Thanks for the positive feedback.
Round 2
Reviewer 1 Report
Comments and Suggestions for Authors
The Ms has improved considerably. As it stands, it provides a more fluid and impartial reading between biological and environmental factors. Table 2 brings relevant information conected to the scientific evidence. Finally, I have only two last suggestions: 1) The term “proximal factor” still bothers me and may raise confusion. Moreover, it appears in the conclusion part only. So I suggest that it could be clarified at some point in the article, previously the conclusion or in the beggining of the conclusion. For instance, it could be included in brackets that this term refers to "a direct and immediate risk contributor". 2) In table 2: just to present the Evidence Strength in perspective of a measuring scale, I would change the line of PFAAs and Vit D. As a result, the table will present the factors from the highest to the lowest evidence.
Author Response
Reviewer 1
The Ms has improved considerably. As it stands, it provides a more fluid and impartial reading between biological and environmental factors. Table 2 brings relevant information connected to the scientific evidence. Finally, I have only two last suggestions: 1) The term “proximal factor” still bothers me and may raise confusion. Moreover, it appears in the conclusion part only. So I suggest that it could be clarified at some point in the article, previously the conclusion, or at the beginning of the conclusion. For instance, it could be included in brackets that this term refers to "a direct and immediate risk contributor". 2) In Table 2, just to present the Evidence Strength in perspective of a measuring scale, I would change the line of PFAAs and Vitamin D. As a result, the table will present the factors from the highest to the lowest evidence.
Responses: Thank you for the valuable guidance. We have added the phrase “(i.e., direct and immediate risk contributors)” immediately following the term “proximal risk factors” in the conclusion section to enhance clarity. We have reordered the rows for “PFAAs” and “PM2.5 Vitamin D” to present the evidence strength in descending order. These revisions address the reviewer’s suggestions while preserving the manuscript’s scientific rigor and overall flow. No additional text modifications were necessary.
Reviewer 2 Report
Comments and Suggestions for Authors
Dear Authors,
Thank you for your revised manuscript and for addressing the comments provided in the previous review. I have carefully examined the changes you made, and I am pleased to confirm that all the suggestions have been adequately incorporated.
The manuscript is now significantly improved and meets the necessary standards for publication. Therefore, I consider the paper ready for acceptance.
Kind regards
Author Response
Reviewer 2
Thank you for your revised manuscript and for addressing the comments provided in the previous review. I have carefully examined the changes you made, and I am pleased to confirm that all the suggestions have been adequately incorporated.
The manuscript is now significantly improved and meets the necessary standards for publication. Therefore, I consider the paper ready for acceptance.
Response: Thank you for your feedback and for taking the time to review the revised manuscript. We are delighted to know that the changes have satisfactorily addressed the previous comments and that the manuscript now meets the standards for publication. We appreciate your support and are grateful for your recommendation for acceptance.
Reviewer 3 Report
Comments and Suggestions for Authors
All the presented points to improve the manuscript was corrected and showed in the revised version.
Author Response
Reviewer 3
All the presented points to improve the manuscript were corrected and shown in the revised version.
Response: Thank you for the feedback. All the suggested improvements have been addressed and incorporated into the revised manuscript.
Reviewer 4 Report
Comments and Suggestions for Authors
Review #2
Dear Authors,
You put a lot of work into preparing the article. However, factual errors prevent it from being published in this version.
1.Various studies have identified potential impacts of smoking on caries development, but the exact processes responsible for the adverse effects of smoking on the development of dental cavities is still not proved.
In your paper you try to prove, analogous to smoking, that unsafe sanitation, handwashing, ambient particulate matter , household air pollution and unsafe water are responsible for ECC.
To do that, you must find potential connections for the disease and its factors. Cite original papers. So far, it is not done.
Try to edit the paper in a way that you have a background for your thesis.
- You repeat the sentence „ For children 0-9 years old, behavioural risk factors such as nutrition-related factors, including wasting underweight, and stunting, known to be associated with ECC”.
I would repeat, the ECC does not exist in children at that age.
After editing the paper, it would be possible to review that.
Author Response
Reviewer 4
You put a lot of work into preparing the article. However, factual errors prevent it from being published in this version.
- Various studies have identified potential impacts of smoking on caries development, but the exact processes responsible for the adverse effects of smoking on the development of dental cavities have not been proven. In your paper, you try to prove, analogous to smoking, that unsafe sanitation, handwashing, ambient particulate matter, household air pollution, and unsafe water are responsible for ECC. To do that, you must find potential connections for the disease and its factors. Cite original papers. So far, it is not done. Try to edit the paper in a way that you have a background for your thesis.
Response: Thank you for your detailed feedback and guidance. We noticed a continued reference to an assumption raised in a prior revision. While we previously deflected from addressing this in the previous response, we recognize the need to clarify this assumption now. This assumption seems to introduce bias to the manuscript review process. This manuscript is not part of any thesis. Three of the four contributing authors are experienced professors of paediatric dentistry with over 15 years of academic and research expertise. In addition, they are globally recognised top researchers on ECC (Zhai L, Kong J, Zhao C, Xu Y, Sang X, Zhu W, Yao N. Global trends and challenges in childhood caries: a 20-year bibliometric review. Transl Pediatr. 2025 Jan 24;14(1):139-152. doi: 10.21037/tp-24-415.). The objective of this work is to generate new insights into the etiology of ECC and stimulate further research in this field.
Regarding the analogy to smoking, we would like to clarify that the paper does not claim that unsafe sanitation, inadequate handwashing, ambient particulate matter, household air pollution, or unsafe water are directly analogous to smoking as causal factors for ECC. Instead, we explored mechanisms by which smoking may influence ECC development. Specifically, the paper discusses how nicotine disrupts ameloblast function, leading to compromised enamel, alters the oral microbiome by suppressing mucosal immunity and promoting Streptococcus mutans colonization, and impairs salivary gland function, resulting in reduced buffering capacity and saliva flow. These pathways collectively lower the threshold for cavitation and accelerate ECC progression in primary teeth. We acknowledge the importance of citing original research to strengthen proposed connections between environmental factors and ECC, and the manuscript cited references 30, 46-50 for this purpose.
- You repeat the sentence „ For children 0-9 years old, behavioural risk factors such as nutrition-related factors, including wasting underweight, and stunting, known to be associated with ECC”. I would repeat, the ECC does not exist in children at that age. After editing the paper, it would be possible to review that.
Response: We fully agree that ECC does not occur in children aged 6–9 years. The referenced age bracket (0–9 years) includes younger children (0–5 years) who are affected by ECC. In addition, the data is about primary teeth, which is the focus of ECC. The sentence referred too in the reviewers’ comment was intended to highlight behavioural risk factors (e.g., nutrition-related factors such as wasting, underweight, and stunting) that are known correlates of ECC, not to suggest that ECC itself occurs in older children. We have opted to retain the sentence and are truly grateful for the insightful comments and thorough review of our manuscript. The quality has improved from the insights shared.